# Diagnosis and Prevalence of Chagas Disease in an Indigenous Population of Colombia

**DOI:** 10.3390/microorganisms10071427

**Published:** 2022-07-14

**Authors:** Simone Kann, Juan Carlos Dib, Andrés Aristizabal, Gustavo Concha Mendoza, Hugo Dario Soto Lacouture, Maria Hartmann, Hagen Frickmann, Lothar Kreienbrock

**Affiliations:** 1Medical Mission Institute, 97074 Wuerzburg, Germany; 2Department of Medicine, Fundación Universidad de Norte, Baranquilla 080001, Colombia; juandib@fspt.co; 3Tropical Health Foundation, Santa Marta 470003, Colombia; anjhoar@gmail.com; 4Organisation Wiwa Yugumaiun Bunkuanarrua Tayrona (OWYBT), Valledupar 200001, Colombia; gustavoconcha16@gmail.com; 5Laboratorio de Entomología, Laboratorio de Salud Pública del Cesar, Valledupar 200001, Colombia; entomologialsp2017@gmail.com; 6Department of Biometry, Epidemiology and Information Processing, WHO Collaborating Centre for Research and Training for Health in the Human-Animal-Environment Interface, University for Veterinary Medicine Hannover, 30559 Hannover, Germany; maria.hartmann@tiho-hannover.de; 7Institute for Medical Microbiology, Virology and Hygiene, University Medicine Rostock, 18057 Rostock, Germany; hagen.frickmann@med.uni-rostock.de; 8Department of Microbiology and Hospital Hygiene, Bundeswehr Hospital, Hamburg, 20359 Hamburg, Germany

**Keywords:** Wiwa, chagas rapid test, serology, chagas-RT-PCR

## Abstract

Chagas disease (CD) is one of the leading neglected tropical diseases. In the Americas, CD is endemic in about 21 countries, but only less than 1% of the patients have access to medical treatment. Indigenous populations are particularly affected because they live in socio-economic and climate conditions that favor CD infections. In this study, diagnostic strategies and regional prevalence of the Chagas disease were assessed. In nine villages of the indigenous tribe Wiwa, 1134 persons were tested with a Chagas-antibody-specific rapid test (RT), two different Chagas-antibody-specific ELISAs and a Chagas-specific real-time polymerase chain reaction. The overall prevalence of CD in the villages was 35.4%, with a variation from 24.9% to 52.8% for the different communities. Rapid tests and ELISAs showed the same results in all cases. The proportion of replication-active infections, defined by positive PCR results, was 8.7%. In conclusion, the assessed indigenous population in Colombia was shown to be severely affected by CD. For a serological diagnosis, one rapid test was shown to be sufficient. Replacements of ELISAs by RT would decrease costs, increase feasibility and would relevantly help detect positive patients, especially if combined with the applied real-time PCR protocol. Real-time PCR can be considered for the detection of acute cases, outbreaks, chronic cases with re-infection/activation, as well as for therapy management and control.

## 1. Introduction

Chagas disease (CD) is endemic in about 21 countries of the Americas. The disease has spread to recently Chagas-free regions, mainly due to migration, and can now also be diagnosed in countries such as the USA, Canada, many European and some African, eastern Mediterranean and western Pacific countries. About 6 million people are infected and 70 million live with a daily risk of infection [1]. Although many efforts have been made to control the disease, and some improvements were achieved, still, less than about 1% of the infected have access to diagnosis and treatment [2]. This causes high morbidity and mortality rates with more than 12,000 deaths per year [1,3,4].

CD is caused by the protozoen parasite *Trypanosoma cruzi* (*T. cruzi*). The parasite can enter the body in various ways; transmission via Triatomines is the most common in the indigenous communities. During their blood meal, they defecate nearby the suction place, and the stool, infected with *T. cruzi*, can enter the body by scratching and/or over mucous membranes. Congenital transmission, infections through blood and/or organ donors, food contamination and laboratory accidents, etc., are other possibilities.

After an acute stage of CD infection, which is associated with unspecific flu-like symptoms in most cases, about 30–40% show disease progression into a chronic stage with complications, such as dilatative cardiomyopathy, going along with heart failure, arrhythmia, sudden heart death, etc., and/or gastrointestinal complaints, such as megacolon, megaesophagus and others [5]. Although transmission of *T. cruzi* by Triatomines is the main route in that area, congenital infections or infections via food contamination, blood donations, organ transplants or accidents in the laboratory are other potential sources. As CD mainly affects young adults in their productive phase, their mortality and morbidity rates affect not only their lives and families but also represent a high economic burden for the countries. The global costs are estimated to be about USD 7.19 billion/year [5]. The health insurance of indigenous customers, however, covers only basic care, diagnostics and treatments, which do not include, e.g., pacemaker interventions or even heart transplants in most cases [6,7]. Therefore, CD in indigenous populations leads to even higher fatality rates than in other populations.

The main problem is that many cases of CD remain undetected, e.g., due to a lack of diagnostic options, awareness and/or missing access to health care [2,8,9,10]. Further, indigenous populations are often neglected and are therefore not included in many statistics, as their living areas are far apart from cities and difficult to reach. Accordingly, higher numbers of infections and death rates than reported are likely. All the efforts and successes listed so far, such as the decrease from 18 million infected in 1991 to fewer than 6 million infected in 2010 [11,12], are therefore without any measurable impact on them and their region.

The indigenous tribe called Wiwa lives in the retracted areas of the Sierra Nevada de Santa Marta in the northeast of Colombia. Already in previous studies, the Wiwa communities have shown to be highly affected by various infectious diseases [13]. As trust could be created and the leadership of the communities asked for advanced support, further villages were examined. To reach these villages, trips of up to 7 h walking were needed. The reasons for their high burden of infectious diseases, and also CD, include their poor socio-economic situation and living conditions (clay huts, palm roofs, etc.), which support the breeding and living conditions of Triatomines, the CD vectors. Although only a few of the more than 130 different Triatomine species transmit the pathogen *Trypanosoma cruzi* (*T. cruzi*) [12], the most important ones are found in the examined zone, such as *Rhodnius prolixus* [14].

To avoid severe complications and cost-intensive therapies, it is most important for the patient to be diagnosed early, ideally in the acute stage of CD, as this is the most promising stage for a cure [15]. Conversely, the access to treatment is difficult. In Colombia, the patients need two different positive serologic test results to be entitled to treatment. If one of the tests shows a discordant result, a third test is required. This diagnostic path was implemented in the Colombian guidelines following the WHO recommendations [16,17]. Unfortunately, two to three different test assays are usually not available in the rural zones of Colombia. Other problems include the costs for the patient, the required infrastructure, such as a laboratory, personnel and the equipment to perform the tests, the complicated process of registration before treatment and others.

Serologic assays only detect patients in their chronic stage, so the best chances of a good therapeutic outcome are diminished. Serologic tests cannot monitor the therapeutic course, success or failure because the antibodies stay positive for many years [18], and changes in their values cannot be used for any interpretation. In addition, the non-pathogenic *T. rangeli* exists sympatrically with *T. cruzi* in the examined region, which leads to false positive results due to cross reactivity in some assays. These serologic limitations can be overcome by the recently described, patent-protected NDO real-time PCR, which can detect acute cases, chronic cases with re-infections or re-activations and which can distinguish between *T. cruzi* and *T. rangeli* [8]. Furthermore, with the NDO real-time PCR, the course of treatment can be monitored, and decisions about therapy failure or success can be made [8]. This approach could also be helpful for CD screenings in pregnant women and newborns. In blood donations or organ transplants, it might improve safety. Additionally, it can be used for mammalian animals and surveillance purposes (e.g., analyzing Triatomines).

The study presented here was performed (1) to define the prevalence of CD in an indigenous population in Colombia, (2) to treat positively tested CD patients within the program, (3) to provide an epidemiologic overview of the villages examined and (4) to answer the question of the feasibility of certain diagnostic tools in a rural Colombian setting.

## 2. Materials and Methods

### 2.1. Study Design and Target Population

The data analyzed were collected from a two-part study. The first part was called “Program against Chagas Disease in the Indigenous Population of Colombia” and took place from July 2017 to March 2019. It included 684 volunteers and is referred to as part I in the following. The second program called “Colombia-Germany research program on diagnostics, research, treatment and prevention of Chagas Disease and Emerging Infectious Diseases in vulnerable groups” started in February 2020, is still ongoing, included 450 volunteers so far and is referred to as part II in this assessment.

In both study parts, the sample collection procedure was the same. Indigenous people from the Wiwa tribe were asked for their voluntary assistance in nine villages: Tezhumake (224) and Cherua (110) (Department Cesar), Ashintukwa (201) and Siminke (149) (Department La Guajira) were analyzed in part I; Ahuyamal (89), Surimena (63), Sabannah de Higuieron (130), Dungakare (68) (Department Cesar) and Potrerito (100) (Department La Guajira) were examined in part II (see Figure 1). In total, data from 1134 Wiwas were collected.

All villages are prone to have a high prevalence of CD and were in the retracted areas of indigenous territories in the northeast of Colombia, the Sierra Nevada de Santa Marta.

### 2.2. Ethical Clearance

Both parts of the study were performed in accordance with the principles of the Declaration of Helsinki. The study providing the data for part I (2017–2019) was approved by the Ethics Committee of Santa Marta, Colombia (Acta No 032018). The study providing the data for part II was approved by the Institutional Ethics Committee for Investigation of Bogota, Colombia (Acta No 2019-4). Written informed consent was obtained from each participant or from the parent or legal guardian of a child prior to participation.

### 2.3. General Screening Design

In all nine communities, the following procedure was performed. Persons of 12 years of age and older first received a Chagas rapid test (RT). If the RT was positive, the serum was taken. The serum was used to additionally perform two different ELISAs and the above-mentioned NDO-RT-PCR.

In volunteers under 12 years of age, the RT, the two ELISAs and the two PCRs were performed in all cases.

All volunteers were informed about their results by a physician and were registered in the official Colombian database of the healthcare provider Dusakawi, guaranteeing all positively tested patients access to treatment. In addition, within the program, there was the offer for all positively tested patients to receive a drug observed treatment, meaning that the treatment was accompanied for a full term of 2 months by a physician. During the treatment phase, all volunteers received physical exams and blood withdrawals on a regular basis to control for possible side effects (liver, kidney values) etc. Women of childbearing age repetitively received pregnancy tests. In addition, any other complaints the doctor was consulted for, whether related to the study or not, were taken care of and documented. A further follow-up on the patients is planned.

### 2.4. Sample Transportation

All samples collected in the villages were cooled in a special sample cooling box at 4 °C (provided by World Courier, Frankfurt, Germany) and transported to the laboratory on the same day. Sera and extracts were stored thereafter at −20 °C. To export the samples to Germany, international guidelines for airfreight sample transport were followed, permissions were obtained from all involved parties, and a permanent cooling chain with dry ice was provided (World Courier, Frankfurt, Germany). After arrival in Germany, sera and extracts were stored at −80 °C until their further use.

### 2.5. Serologic Methods

All collected samples were tested with a Chagas Rapid Test (RT) (Chagas AB Rapid, Standard Diagnostics Inc. Bioline, Bogota, Colombia) and two different ELISAs (Chagatest ELISA recombinant v. 4.0 Wiener Lab, ELISA-Recombinante and Chagas IgG ELISA Lisado IBL, Wiener Lab, ELISA-Lisado), following the manufacturers’ protocols. The choice of the two different ELISAs was performed according to the Colombian guidelines for the diagnosis of CD, which are based on the WHO recommendations.

### 2.6. DNA Extractions

Nucleic acid extractions were carried out from the serum following the instructions of the manufacturers’ protocols of the RTP Pathogen Kit (Invitek Molecular GmbH, Berlin, Germany) with the samples from part I and of the MagaBio plus Virus DNA/RNA purification Kit version 2 (Hangzhou Bioer Technology Co., Ltd., Hangzhou, China) with the samples from part II. The extraction and eluate volumes were in a comparable range over the compared extraction schemes with 200 µL and 60 µL for the RTP assay and 300 µL and 80 µL for the MagaBio assay, respectively. Previous in-house evaluation indicated that both extraction approaches yield comparable results.

### 2.7. Real-Time Polymerase Chain Reaction (RT-PCR)

In part I, T. cruzi/T. rangeli-specific kDNA and T. cruzi-specific NDO-PCR were conducted with two runs each in parallel. The protocols were used as published previously [8]. As the analysis showed the superiority of the NDO-PCR, a basic discordance was seen in only 0.4% of the runs, and as the NDO-PCR became commercially available (TibMolBiol/Roche, Berlin, Germany: T. cruzi LightMix®, Ref 53-0755-96, PhHV Extraction Ctrl. Ref. 66-0901-96, Lyophilized 1-step RT-PCR Polymerase Mix, Cat-No 90-9999-96), we performed only one NDO-PCR run in part II. To compare these results with part I, we used one positive run only for the assessments. As a minor modification compared to the previous NDO-RT-PCR protocol, the probe sequence was slightly altered to 5′-TCG + AACCCC + ACCTCC-3′, the “+” symbol marking the locked nucleic acid (LNA) bases included to alter the annealing temperature. A synthetically produced positive control and the above-mentioned T. cruzi strain Tulahuen-based positive control were used in parallel. All runs were performed on a RotorGene Q cycler (Qiagen, Hilden, Germany).

### 2.8. Disease Classification

Acute and newly infected cases were defined by the direct detection of the parasite in the blood [19], i.e., by a positive PCR run.

As two different serological techniques are recommended by the WHO and demanded by the Colombian guidelines to entitle the patient to treatment, we applied two different ELISAs. In addition, we performed a Chagas antibody-specific RT.

According to the results, patients testing positive in the two ELISAs (and the RT) and negative in all PCR runs were classified as chronically infected and those testing positive in the two ELISAs (and the RT) and positive in at least one PCR run as acutely re-infected, re-activated and/or at an (early) chronic stage/late acute stage of CD infection.

### 2.9. Data Management and Statistical Analyses

The obtained data were transferred into the SAS statistical analysis program (version 9.4 TS level 1M5, SAS Institute Inc., Cary, NC, USA) for data description and testing of statistical significance. Data analyses were performed both on the Chagas disease status as defined above, as well as in a dichotomous way, comparing the outcomes of Chagas-negative and -positive results only. A multi-factorial logistic regression was performed to control for possible confounding. For this analysis, age was stratified into five age classes. As an exploratory error level, 5% was accepted.

## 3. Results

### 3.1. Demographic Data of the Study Population

The total number of collected blood samples was 1134 collected from nine Wiwa villages. In all villages, the majority of inhabitants were screened (sample population n/participation rate in %), namely Tezhumake 224/89.6%, Seminke 149/96.1%, Cherua 110/91.6%, Ashintukwa 201/80.4%, Ahuyamal 89/90.6%, Surimena 63/92.6%, Sabannah de Higuieron 130/95.6%, Dungkare 68/89.2% and Potrerito 100/90.1%.

Overall, 577 (50.9%) samples were from females and 557 (49.1) from male volunteers. In total, 353 (31.1%) persons were younger than 12 years, 777 (68.5%) were 12 years and older, and in 4 (0.4%) cases, the age was unknown. The median age was 18 years (minimum, maximum 1, 90). The median age of the volunteers under 12 years was 7.1 years; the median age of the participants 12 years of age and above was 25.3 years, with only minor differences between male and female participants.

### 3.2. Rapid Test (RT) and ELISA Results

The RT was negative in 732 cases and positive in 402 cases, i.e., the total RT prevalence in all villages screened was 35.45%. This result was concordantly found with and between both ELISAs in both study parts I and II in all 1134 participants.

### 3.3. PCR Results and Combined Chagas Status of Participants

PCR runs were performed in duplicate for NDO- and kDNA-PCR, resulting in a total of four runs in part I and one run only with the NDO-PCR in part II.

Using these results, different diagnostic patterns occur, namely, three results from serology (RT, ELISA-Lisado, ELISA-Recombinante) and two results from NDO-PCR in part I and one result from part II. Overall, this resulted in a Chagas status definition outlined in Table 1.

Of the 1134 subjects tested, 64.37% showed a negative test result for all tests (serology and PCR). Positive test results can be divided into three categories, namely “sero-positive and NDO-positive” at 8.55%, “sero-positive and NDO-negative” at 26.89% and “sero-negative and NDO-positive” at 0.18%.

This means we found two cases that were acutely infected. This was one girl from Ahuyamal, 7 years old, and one girl from Sabannah de Higuieron, 4 years of age.

### 3.4. CD Diagnosis and Prevalence by Demographic Characteristics

The outcome described in Table 1 was related to the general demographics of the entire study population of all nine villages. Table 2 and Table 3 and Figure 2 show these outcome results stratified by age distribution and village, respectively.

Chagas status by sex shows a moderate, statistically significant effect in the general Chi^2^ test (*p* = 0.0219), which disappears when all positive findings are aggregated into one category (*p* = 0.2953).

Age distributions in all classes of the Chagas status show a strong, statistically significant effect in an ANOVA model (global F-test, *p* < 0.0001). Looking into the least-square means for the group effects, this significance level holds true for both tests between the “negative” and the “sero-positive and NDO-positive” groups, as well as the “sero-positive and NDO- negative” groups (*t*-test, *p* < 0.0001). The test for “sero-negative and NDO-positive” does not show a significant result (*t*-test, *p* = 0.3133).

CD prevalence ranged from 24.9% in Ashintukwa to 52.8% found in Ahuyamal. This resulted in a statistically significant effect in the Chi^2^ test (*p* = <0.0001). The ranking of prevalence was: Ahuyamal, Surimena, Cherua, Sabannah de Higuieron, Tezhumake, Seminke, Dungakare, Potrerito and Ashintukwa.

Taking into account a possible confounding resulting from unbalanced sampling in the villages, all three demographic variables were introduced into a multi-factorial logistic regression model with the dichotomous Chagas status as the outcome (Table 4).

Within the multi-factorial approach, sex does not show a significant effect on Chagas prevalence, while a large age effect is shown. The older the population, the higher the risk of a positive Chagas prevalence. In contrast to the reference group of children younger than six years, the Chagas odds are enlarged more than 100-fold. In addition, the big variety in prevalence between different villages is shown, as well with an extended statistical significance (see Table 5).

## 4. Discussion

The indigenous tribe called Wiwa lives in the retracted areas of the Sierra Nevada de Santa Marta, in the northeast of Colombia. Due to poor living conditions, little access to surveillance and healthcare programs, the successes and innovations with regard to CD management have not yet reached them. Instead, they suffer from high prevalence of the disease, ranging from 24.9% in Ashintukwa up to 52.8% in Ahuyamal, presenting an overall prevalence in the region of 35.4% in nine villages. According to the Pan-American Health Organization (PAHO), the highest seroprevalence in the world can be found in Bolivia (average of 6.8%) [1]. The Bolivians have found up to 50% seropositivity in some communities [20,21]. Compared to this, the examined regions in the Sierra Nevada de Santa Marta are even slightly more affected than the most affected villages in Bolivia (50% vs. 53%). The overall numbers given for Colombia show that approximately 10% of the population is supposed to live in endemic areas, and the national prevalence is estimated to be at nearly 1% [22]. However, this might need to be reconsidered after including the data from the indigenous populations.

Many efforts have been made in various countries to improve the situation, comprising education, housing enhancements, insecticide sprayings, blood donor screenings, etc. However, in the Wiwa regions, only a few of these actions have taken place. Considering that studies on the global burden suggested in 2010 that CD was responsible for 550,000 DALYs (premature mortality and non-fatal health loss included) [23,24] and documented this high burden in the indigenous communities, the countermeasures are urgently needed.

The WHO Expert Committee states that the geographic distribution of Chagas is driven by the distribution of the vector species and that vector-borne transmission is limited to the Americas, between 40° N and 45° S latitude and below 1500 m elevation [1,23]. However, we found high numbers of infections in villages above 1500 m elevation, such as in Surimena (1860 m) and in Dungakare (1680 m), and we also found the transmitting triatomines there, such as, e.g., *Rhodnius prolixus*, suggesting that this statement needs to be corrected.

As the examined region can be considered a high endemic area, mostly chronic cases were detected (35.4%). Interestingly, from these serologically positive cases, 8.6%, were also PCR positive, showing that *T. cruzi* was circulating in their blood. This high proportion and also the two fresh infections identified (0.18%), as well as the age-related dependence of infection on an exponential scale, clearly indicated the high activity of *T. cruzi* in the region.

To make the detection of *T. cruzi* easier, it should be discussed whether the recommendation by the WHO [25], requiring two different positive ELISAs to entitle the patient to treatment, can be improved. The study indicated 100% concordant results for the rapid tests (RT) compared to both ELISAs, which showed identical results in both assays in 100% of the cases as well. Performing ELISA assays is technically demanding in rural settings. Laboratories, equipment, personnel and money are often missing; one or even two different ELISAs or alternative serologic tests are frequently not available. In contrast, the applied RT is very cheap, easy to perform and useful in the field. As described previously by various colleagues [26,27,28], the data presented herein also demonstrated that the RT could be a good and useful replacement for technically demanding and time-consuming ELISAs or other technically demanding serological approaches. It would tremendously raise the detection rate of people being infected, increasing their chances of the benefit of receiving treatment. Based on the experience of the study described here and the observations made by others [29,30], it should be considered whether the WHO recommendation [25] should be changed to allow a Chagas disease diagnosis for chronic cases to be based on an RT result.

As traditional Chagas serologies are indirect diagnostic approaches focusing on antibodies, which persist for years or even lifelong [18,21], these methods are difficult to use for monitoring the therapy and the estimation of therapeutic success or failure. With the recently described *T. cruzi*-specific NDO-RT-PCR, there is a tool available now that can reliably close this gap. As the PCR is very specific, it does not cross-react either with the apathogenic *T. rangeli* or other relevant pathogens in the area, such as *Leishmania* spp. [8,31]. As acute cases benefit from early treatment, the PCR can be used for this indication as well. As it measures *T. cruzi* DNA directly in the blood, re-activated, re-infected cases can also be found, and even newborns with CD-positive mothers could be screened. The also applied and frequently used kDNA-PCR assay, in contrast, confirmed its specificity limitations once more, as high rates of false positive results due to cross-reactions with *T. rangeli* DNA [8,9,10] were seen in this study as well. Another limitation of the kDNA-PCR is, thant quantification is often not possible.

As a limitation of the study, it needs to be disclosed that in part I of the study, two runs of the NDO-PCR were performed; in part II, only one run was performed. The practical relevance is, however, low, as the two runs in part I showed a 99.6% concordance of the results, so one run can be considered as reliable and sufficient. Further, follow-up data on the population would be favorable, and their collection is in progress. It needs to be mentioned that positive cases in PCR and serology could also be classified as acute cases, infected with certain Tc-DTUs that persist even after treatment. Therefore, it would have been favorable if sequencing could have been performed; however, due to the limitations of the budget, this was not possible within this study.

The high prevalence and little access to diagnostic and medical care clearly indicated that the countermeasures to avoid the further spread of the disease and to improve the situation are urgently needed. This includes, e.g., surveillance data, an improvement of infrastructure, accessibility of treatment and good diagnostics.

## 5. Conclusions

The assessment indicated very high seroprevalence rates of CD in the examined regions of the Sierra Nevada de Santa Marta in the local indigenous population. Accordingly, the area can be considered very highly endemic. To improve the screening for CD, a replacement of the two recommended serologic tests (ELISAs) by reliable rapid tests should be considered by the WHO and the providers of local guidelines. For the detection of acute cases and to monitor the course of the therapy, the NDO-RT-PCR could be applied if PCR laboratories are available.

## Figures and Tables

**Figure 1 microorganisms-10-01427-f001:**
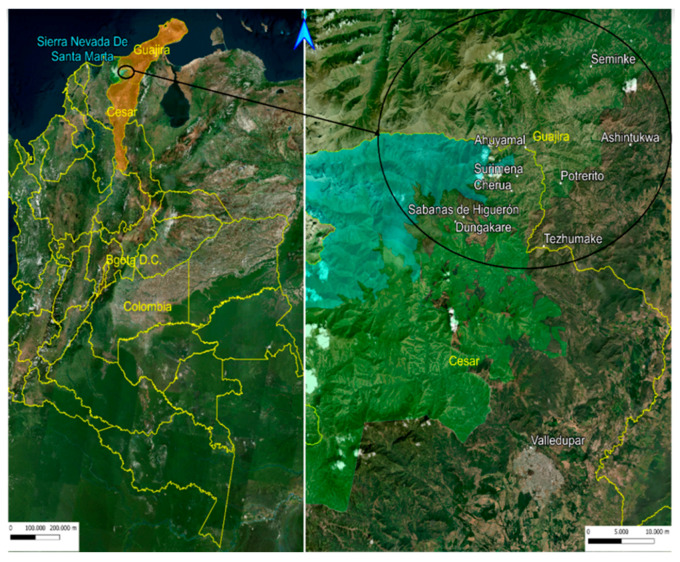
Maps of Colombia and Wiwa indigenous settlements, map made with Qgis Hannover 3.16 (https://wwwqgis.org/es/site/) modified by H. Soto, 12 July 2022.

**Figure 2 microorganisms-10-01427-f002:**
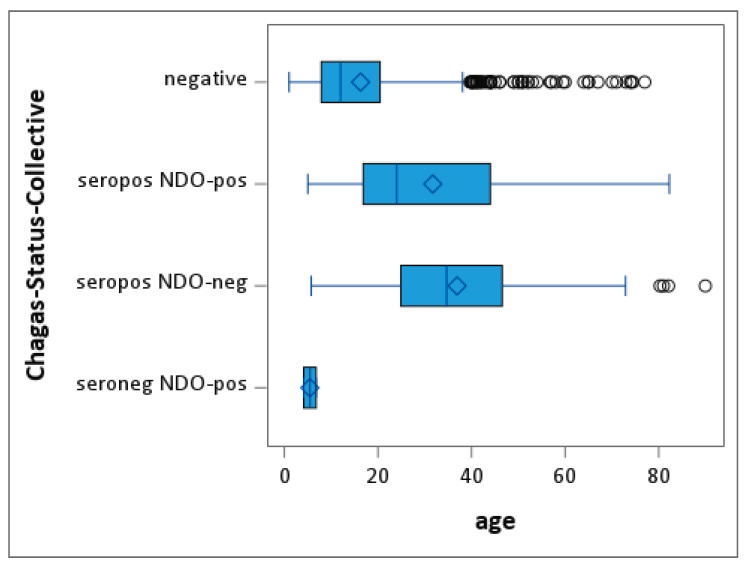
Box plots of age distribution by Chagas status in the entire study population (*n* = 1134).

**Table 1 microorganisms-10-01427-t001:** Combined presentation of serological and PCR results of Chagas diagnostics in the entire study population (*n* = 1134).

Chagas Status	Pattern Serology *	Part	*n*	%
negative	0..	I	200	17.64
negative	0..	II	164	14.46
negative	000	I	264	23.28
negative	000	II	102	8.99
All			730	64.37
Sero-positive/NDO-positive	111	I	57	5.03
Sero-positive/NDO-positive	111	II	40	3.53
All			97	8.56
Sero-Positive/NDO-negative	111	I	163	14.37
Sero-Positive/NDO-negative	111	II	142	12.52
All			305	26.89
Sero-negative/NDO-positive	000	I	0	0
Sero-negative / NDO-positive	000	II	2	0.18
All			2	0.18
Total			1134	100.0

* serological pattern 0 = negative, 1 = positive, 0 = test not applied is from three tests, namely rapid test, ELISA-lisado and ELISA-recombinante.

**Table 2 microorganisms-10-01427-t002:** Chagas status in the entire study population by sex (*n* = 1134).

Chagas Status	Sex				All
	Female		Male		
	*n*	Percent	*n*	Percent	N
negative	363	62.91	367	65.89	730
seropos NDO-pos	40	6.93	57	10.23	97
seropos NDO-neg	172	29.81	133	23.88	305
seroneg NDO-pos	2	0.35	0	0	2
All	577	100.00	557	100.00	1.134

**Table 3 microorganisms-10-01427-t003:** Measures of age distribution by Chagas status in the entire study population (*n* = 1134).

Chagas Status	*n*	Mean	Median	STD	CV	Min	5% Perc	95% Perc	Max	Missing
negative	727	16.28	12.00	13.01	79.93	1.00	4.00	44.00	77.00	3
seropos NDO-pos	97	31.71	24.00	19.67	62.05	5.00	8.64	77.03	82.26	0
seropos NDO-neg	304	36.90	34.68	15.60	42.29	5.74	15.21	64.27	89.95	1
seroneg NDO-pos	2	5.45	5.45	1.88	34.55	4.12	4.12	6.78	6.78	0
ALL	1130	23.13	17.96	17.18	74.29	1.00	4.00	59.22	89.95	4

**Table 4 microorganisms-10-01427-t004:** Chagas status in the entire study population by village (*n* = 1134).

Village	Chagas Status								All
	Negative		Seropos NDO-Pos		Seropos NDO-Neg		Seroneg NDO-Pos		
	*n*	Percent	*n*	Percent	*n*	Percent	*n*	Percent	*n*
Seminke	101	67.8	18	12.1	30	20.1	0	0	149
Ashintukwa	151	75.1	17	8.5	33	16.4	0	0	201
Cherua	66	60.0	5	4.5	39	35.5	0	0	110
Tezhumake	146	65.2	17	7.6	61	27.2	0	0	224
Ahuyamal	42	47.2	13	14.6	33	37.1	1	1.1	89
Dungakare	46	67.6	3	4.4	19	27.9	0	0	68
Potrerito	72	72.0	11	11.0	17	17.0	0	0	100
Sabana de Higuerón	72	55.4	7	5.4	50	38.5	1	0.8	130
Surimena	34	54.0	6	9.5	23	36.5	0.	0.	63
All	730	64.4	97	8.6	305	26.9	2	0.2	1134

**Table 5 microorganisms-10-01427-t005:** Adjusted odds ratio estimates from multi-factorial logistic regression in the entire study population (*n* = 1134).

Risk Category	Odds Ratio Point Estimate	95% Wald Confidence Limits	
		Lower Bound	Upper Bound
sex (reference category: female): Wald’s Chi^2^-*p* = 0.6805			
male	0.937	0.688	1.277
age (reference category: <6 years): Wald’s Chi^2^-*p* < 0.0001			
6–<12	1.948	0.535	7.091
12–<18	10.305	3.063	34.670
18–<24	34.327	10.152	116.069
≥24	111.826	33.947	368.371
village (reference category: Ashintukwa): Wald’s Chi^2^-*p* < 0.0001			
Ahuyamal	9.721	4.855	19.462
Surimena	4.055	1.881	8.742
Cherua	3.081	1.642	5.783
Sabannah de Higuieron	3.005	1.688	5.350
Tezhumake	2.860	1.695	4.824
Seminke	2.003	1.129	3.552
Dungakare	1.009	0.507	2.009
Potrerito	0.823	0.445	1.523

## Data Availability

Not applicable.

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
