# Peer review of "Diagnosis and Prevalence of Chagas Disease in an Indigenous Population of Colombia"

_microorganisms, 2022, doi:10.3390/microorganisms10071427_

Round 1

Reviewer 1 Report

Thank you for the opportunity to review the Paper for submission, “Diagnosis and Prevalence of Chagas Disease in an Indigenous Population of Columbia”. The authors studied a volunteer sample of the Columbian Wiwa tribe, a group living in a high incidence area of Chagas Disease. The listed study aims were to: 1- define prevalence of Chagas disease in the sample,2- assess options for referral for treatment, 3-perform epidemiologic overview, and 4-assess diagnostic tools for Chagas disease.

1100 people from nine villages of the Wiwa tribe were sampled. About 90% of the population participated. A rapid Chagas test was done, if positive it was followed by two ELISA tests and 2 NDO real time PCR tests to assess for active disease. Samples were sent to Germany for processing. The authors sorted the study population into groups by the pattern of testing: those negative for Chagas, those with positive or negative  serologies/ positive or negative real time PCR. Positive PCR was thought to indicate actively replicating  disease. The researchers also had access to the participants’ sex, age and village lived in. Sex did not impact infection status but age and village of origin did.  The percentage of positive PCR patients was felt to reflect a realistic percentage of those with active disease. There was no clinical screening for clinical disease manifestations.

This research paper contributes to the Chagas literature and has the potential to impact care of remote populations afflicted with Chagas. The team had excellent recruitment to screening suggesting a thorough sampling of the population .The application of an intensive laboratory screening in a remote area is a tremendous undertaking. The PCR technology has great potential for targeting candidates for treatment.

I recommend publication in Microorganism . I have suggestions for modifications prior to publishing.

1 Introduction- I agree with offering readers review on the epidemiology and natural history of Chagas given how few scientists/physicians, even in the infectious disease world have much knowledge on the topic. I suggest moving the paragraph starting on line 66 “ after an acute stage of CD…” to line 52 . this would organize the introduction starting with generalizations about CD then getting to the specifics of the research question.

2 Introduction – consider adding a reference Bauer H Trop. Med. Infect. Dis. 2022, 7, 109 “Prevalence of common diseases in indigenous peoples in Colombia”. This ties this research to other studies of the Wiwa. I would like a little more background on the choice of this community. Did the authors have a prior relationship? Accessibility? Leadership open to medical screening?

3 Introduction- Expand paragraph on testing starting on line 90 . The NDO PCR could be a breakthrough. Challenges in testing and diagnosis are a huge barrier to care and treatment. Developing a readily applied diagnostic algorithm is central to improving care for this neglected disease.

4 Introduction – paragraph starting in line 100 – After reading through the paper, I think the listed specific aims should be modified. Aim 1  was well addressed. Aim 2  was not addressed , Aim 3 seems too broad since there wasn’t a random sample and extrapolation   Aim 4 really was to assess “feasibility” of diagnostic tools since diagnosis was not undertaken as part of a more comprehensive plan of disease management. Could I suggest that Aims 2 and 3  be put at the end as a future direction and Aim 4 be modified to feasibility?

 5 Methods :Given this is a vulnerable population, a description of how positive tests were handled would be useful. Were subjects notified of their results? Were there options for therapy and follow up?

6 analysis- Is there a biological reason to think that males and females would be differentially affected by CD? Is a separate table 2 necessary ? (Age makes more sense for acute/ chronic infection.) Are the villages different in some way that might explain the differences in prevalence?

7 Language changes

Line 66 change “flue” to “flu”

Line 67 correct to “symptoms

Line 296  change “novalties” to “innovations”

Author Response

Thank you for the opportunity to review the Paper for submission, “Diagnosis and Prevalence of Chagas Disease in an Indigenous Population of Columbia”. The authors studied a volunteer sample of the Columbian Wiwa tribe, a group living in a high incidence area of Chagas Disease. The listed study aims were to: 1- define prevalence of Chagas disease in the sample,2- assess options for referral for treatment, 3-perform epidemiologic overview, and 4-assess diagnostic tools for Chagas disease.

1100 people from nine villages of the Wiwa tribe were sampled. About 90% of the population participated. A rapid Chagas test was done, if positive it was followed by two ELISA tests and 2 NDO real time PCR tests to assess for active disease. Samples were sent to Germany for processing. The authors sorted the study population into groups by the pattern of testing: those negative for Chagas, those with positive or negative  serologies/ positive or negative real time PCR. Positive PCR was thought to indicate actively replicating  disease. The researchers also had access to the participants’ sex, age and village lived in. Sex did not impact infection status but age and village of origin did.  The percentage of positive PCR patients was felt to reflect a realistic percentage of those with active disease. There was no clinical screening for clinical disease manifestations.

This research paper contributes to the Chagas literature and has the potential to impact care of remote populations afflicted with Chagas. The team had excellent recruitment to screening suggesting a thorough sampling of the population .The application of an intensive laboratory screening in a remote area is a tremendous undertaking. The PCR technology has great potential for targeting candidates for treatment.

I recommend publication in Microorganism . I have suggestions for modifications prior to publishing.

1 Introduction- I agree with offering readers review on the epidemiology and natural history of Chagas given how few scientists/physicians, even in the infectious disease world have much knowledge on the topic. I suggest moving the paragraph starting on line 66 “ after an acute stage of CD…” to line 52 . this would organize the introduction starting with generalizations about CD then getting to the specifics of the research question.

At first I would like to thank you for your kind words and really do appreciate!

Yes, we re-organized the section according to your suggestion, so that the readers flow is improved.

2 Introduction – consider adding a reference Bauer H Trop. Med. Infect. Dis. 2022, 7, 109 “Prevalence of common diseases in indigenous peoples in Colombia”. This ties this research to other studies of the Wiwa. I would like a little more background on the choice of this community. Did the authors have a prior relationship? Accessibility? Leadership open to medical screening?

We explained this in more detail and added the following sentence in the introduction part:

“Already in previous studies, the Wiwa communities have shown to be highly affected by various infectious diseases [11]. As trust could be created and the leadership of the communities asked for advanced support, further villages were examined. To reach this villages, trips up to 7 hours walking were needed. Reasons for their high burden of infectious diseases and also CD are:…”

3 Introduction- Expand paragraph on testing starting on line 90 . The NDO PCR could be a breakthrough. Challenges in testing and diagnosis are a huge barrier to care and treatment. Developing a readily applied diagnostic algorithm is central to improving care for this neglected disease.

Thank you for that hint. We have expanded this part and added the following sentences and a citation:

“Furthermore, with the NDO real-Time PCR the course of the treatment can be monitored and decisions about a therapy failure or success, can be made (8). This approach could also be helpful for CD screenings in pregnant women and newborns. In blood donations or organs transplants it might raise the safety. Also for mammalian animals and surveillance purposes (e.g., analyzing Triatomines), it can be used.

4 Introduction – paragraph starting in line 100 – After reading through the paper, I think the listed specific aims should be modified. Aim 1  was well addressed. Aim 2  was not addressed , Aim 3 seems too broad since there wasn’t a random sample and extrapolation   Aim 4 really was to assess “feasibility” of diagnostic tools since diagnosis was not undertaken as part of a more comprehensive plan of disease management. Could I suggest that Aims 2 and 3  be put at the end as a future direction and Aim 4 be modified to feasibility?

We agree, that the aims could be formulated more clearly and rephrased them accordingly. In addition, we added aims 2 and 3 as future research directions in the discussion.

The text in the introduction is now: “The study presented here was performed 1) to define the prevalence of CD in an indigenous population in Colombia, 2) to treat positively tested CD patients within the program, 3) to provide an epidemiologic overview of the villages examined, and 4) to answer the question on feasibility of certain diagnostic tools in a rural Colombian setting.”

The following text was added in the discussion: “The high prevalence and the little access to diagnostic and medical care impressively indicate that countermeasures to avoid the further spread of the disease and to improve the situation are urgently needed. This includes e.g. the generation of more surveillance data, an improvement of infrastructure, accessibility of treatment and good diagnostics.”

 5 Methods :Given this is a vulnerable population, a description of how positive tests were handled would be useful. Were subjects notified of their results? Were there options for therapy and follow up?

Yes, we had already mentioned this in the method section, however, we have now added some more information, so that the chapter is now:

“All volunteers were informed about their results by a physician and registered in the official Colombian data base of the health care provider Dusakawi, guaranteeing all positively tested patients access to treatment. In addition, within the program, there was the offer for all positively tested patients to receive a drug observed treatment, meaning that the treatment was accompanied for the full term of 2 months by a physician. During the treatment phase, all volunteers received physical exams and blood withdrawals on a regular base to control possible side effects (liver, kidney values, etc.). Women in childbearing age repetitively received pregnancy tests. In addition, all other occurring complaints, if related to the study or not, were taken care of and documented. A further follow up on the patients is planned.“

6 analysis- Is there a biological reason to think that males and females would be differentially affected by CD? Is a separate table 2 necessary ? (Age makes more sense for acute/ chronic infection.) Are the villages different in some way that might explain the differences in prevalence?

We agree, that there is general no biological reason for assuming a sex-specific association. However, from an epidemiological perspective it may assumed, that men and women have different social habits and these may linked to the exposure. Therefore sex was tested and incorporated into the multi-factorial models. Finally, no significance was observed if all positive findings were aggregated into one category. To demonstrate this process we therefore like to keep table 2.

7 Language changes

Line 66 change “flue” to “flu”

Line 67 correct to “symptoms

Line 296  change “novalties” to “innovations”

Thank you very much, we have corrected the typing mistakes.

Reviewer 2 Report

The article entitled "Diagnosis and Prevalence of Chagas Disease in an Indigenous population of Colombia", submitted to the journal Microorganism by S. Kann et al, is an interesting study on the prevalence and epidemiology of Chagas Disease in indigenous population in Colombia 

The paper describes the high prevalence of Chagas disease in the indigenous Winwa population in Colombia. The paper compares data from rapid diagnostic tests with molecular diagnostic systems that discriminate between T. rangeli and T. cruzi parasitism.

The data are well presented, relevant epidemiological data with parasitism rates comparable to those recently published in the indigenous population of neighbouring Central American countries. It would have been important to provide a geographic-ecological description of the area occupied by this population and perhaps a map of the region in question.

Otherwise it is an interesting work and should be published.

Reviewer 3 Report

General Comments

The manuscript evaluated strategies of diagnosis and the prevalence of Chagas disease in nine villages of the indigenous tribe Wiwa in the northeast Colombia. A total of 1,134 individuals were tested with a Chagas-antibody-specific rapid test (RT - Bioline), two different Chagas-antibody-specific ELISAs and a Chagas-specific real-time polymerase chain reaction. The overall prevalence of CD in the villages was 35.4%, with a variation from 24.9% to 52.8% for the different communities. Interestingly, Rapid tests and ELISAs showed the same results in all cases, which represents an important contribution of the study, what indicates that for the serological diagnosis, one rapid test was shown to be sufficient for CD diagnosis contrarily to the current recommendation of WHO (1991), especially if combined with the applied real-time PCR protocol (Kan et al., 2020). The authors established a particular Chagas Disease classification taking into account the specific and sensibility of RT-PCR technique used. The proportion of replication-active infections, defined by positive PCR results, was 8.7%. In conclusion, the indigenous population evaluated is severely affected by CD. In addition, the authors suggest that Real-time PCR can be considered for the detection of acute cases, outbreaks, chronic cases with re-infection/activation as well as for therapy management and control.

The study is relevant and its contributions are important. However, some comments and suggestions for the authors follow below in order to clarify some aspects and improve the quality of the manuscript before publication.

Introduction

Authors should include the etiology of Chagas disease at the beginning of the Introduction section.

Ref 1? Line 48:

Line 51: Include the citation of Rassi et al., 2010

Line 58:  I suggest changing Reference 7 by DNDi 2020 (DNDi, Drugs for Neglected Disease initiative, 2020. Q&A COVID-19 and Chagas Disease. https://www.coalicionchagas.org/es/news-article/-/asset_publisher/hJnt8AyJM2Af/content/preguntas-y-respuestas-sobre-covid-19-y-chagas

, which is more recent and actual.

Line 65: Authors should cite a reference after R. prolixus?

Lines 83 – 85: Authors should cite a Ref for this phrase.

Line 85: WHO recommendation is this reference? WORLD HEALTH ORGANIZATION - Control of Chagas Disease. Wld Hlth Org. techn. Rep. Ser., (811): 38-47, 1991. Cite this reference.

Lines 90 – 93: This phrase is not correct because the diagnosis of the acute infection may be performed by serological tests since the first month of infection when IgG is present, associated or not to IgM. Moreover, the acute phase last for approximately four months when IgG is expected to be present in high concentration. PRATA, 2001. RASSI et al., 2010.

The negativation of the serological tests in patients infected with T. cruzi and treated occurs when they are cured (WHO, 2002). This process is slow (several years, decades in chronic infections), but occurs. Besides, the progressive drop of the serological tests after treatment is indicative of therapeutic efficacy (Rassi and Luquetti et al.; 2003). Rassi, A., Luquetti AO. 2003. Specific treatment for Trypanosoma cruzi infection (Chagas disease). In: American trypanosomiasis. Tyler, K.M., Miles, M.A., Kluwer Academic Publishers, Boston, USA. pp. 117-125.

See these references and cut or change this phrase.

VIOTTI et al., 2011; Sguassero et al., 2015; 2018; Krettli , 2009;

Materials and Methods

Line 143: Authors should write what want to say what “drug observed treatment” exactly means. This is an error of expression?

Lines 196 – 197: The Disease Classification used by the authors is different than the classic literature that considers an acute case the presence of the parasite detected by the examination of the blood sample in the coverslip observed in 40X microscope objective, associated or not to IgM presence in the serological test plus symptomatology such the signs of parasite T. cruzi entry besides several others not specific of CD (Luquetti, AO and Schmunis, GA Diagnosis of Trypanosoma cruzi infection. In: Telleria Jenny and Michel Tibayrenc. (Org.). American Trypanosomiasis Chagas Disease One Hundred Years of Research. 2ed.Oxford: ELSEVIER, 2017, p. 687-722).

Again this phrase:  “The last group was interpreted as acutely re-infected, re-activated, and/or as an (early) chronic stage of CD infection.” is different than the classic literature because there is numerous publications that show PCR positive in chronic cases of Chagas disease even after etiological treatment, including in other regions or countries where TcI DTU of T. cruzi is present, as happens in Mexico.  Differently, in regions where TcII, TcV and TcVI the positivity of PCR, is higher as well as the virulence of the parasites and positivity of other parasitological methods of diagnosis.

I understand that this Disease Classification is based on the high credibility of the authors in the RT-PCR performance employed in this study. However, this option needs to be clearly and briefly explained and justified. 

Results

Table 1: % of line 8 of this Table of 8.55 must be changed for 8.56.

Line 254: 8.55% must be changed for 8.56.

Table 2: Take care with the format, particularly of the columns. Signalize with different markers or symbols the statistically significant differences.

Table 4: The alignment of the columns should be ameliorated. I suggest yet the inclusion of a column with the addition of the percentage of reactive cases (serop NDO-neg + serop NDO-pos) for showing easily the situation of all villages.

Discussion

Lines 352 – 355: The cross-reaction of kDNA-PCR between T. cruzi and T. rangeli as well as with Leishmania sp is not a problem today because there are specific primers that avoid this situation. The critical limitation of this procedure is that the kDNA is not quantified and consequently the parasite load is not determined.

Conclusions

Lines 366-367:  The phrase “Countermeasures to 366 avoid the further spread of the disease and to improve the situation are urgently needed” is not a conclusion, but an advertisement or recommendation that should be included in the final of the Discussion section.

Language:

A careful English revision is necessary.

There are some edition errors. Ex. favorale that needs to be replaced by favorable. Some words were written in Spanish.

Author Response

The manuscript evaluated strategies of diagnosis and the prevalence of Chagas disease in nine villages of the indigenous tribe Wiwa in the northeast Colombia. A total of 1,134 individuals were tested with a Chagas-antibody-specific rapid test (RT - Bioline), two different Chagas-antibody-specific ELISAs and a Chagas-specific real-time polymerase chain reaction. The overall prevalence of CD in the villages was 35.4%, with a variation from 24.9% to 52.8% for the different communities. Interestingly, Rapid tests and ELISAs showed the same results in all cases, which represents an important contribution of the study, what indicates that for the serological diagnosis, one rapid test was shown to be sufficient for CD diagnosis contrarily to the current recommendation of WHO (1991), especially if combined with the applied real-time PCR protocol (Kan et al., 2020). The authors established a particular Chagas Disease classification taking into account the specific and sensibility of RT-PCR technique used. The proportion of replication-active infections, defined by positive PCR results, was 8.7%. In conclusion, the indigenous population evaluated is severely affected by CD. In addition, the authors suggest that Real-time PCR can be considered for the detection of acute cases, outbreaks, chronic cases with re-infection/activation as well as for therapy management and control.

The study is relevant and its contributions are important. However, some comments and suggestions for the authors follow below in order to clarify some aspects and improve the quality of the manuscript before publication.

Introduction

Authors should include the etiology of Chagas disease at the beginning of the Introduction section.

Thank you for that hint. We added an etiology section: “CD is caused by the protozoan parasite Trypanosoma cruzi (T. cruzi). The parasite can enter the body by various ways. The transmission via triatomines is most common in the indigenous communities. During their blood meal, they defecate nearby the suction place and stool, infected with T. cruzi, can enter the body by scratching and/or over mucous membranes. Congenital transmission, infections by blood and/or organ donors, food contaminations and laboratory accidents, etc. are other possibilities.”

Ref 1? Line 48:

Line 51: Include the citation of Rassi et al., 2010

Line 58:  I suggest changing Reference 7 by DNDi 2020 (DNDi, Drugs for Neglected Disease initiative, 2020. Q&A COVID-19 and Chagas Disease. https://www.coalicionchagas.org/es/news-article/-/asset_publisher/hJnt8AyJM2Af/content/preguntas-y-respuestas-sobre-covid-19-y-chagas, which is more recent and actual.

Line 65: Authors should cite a reference after R. prolixus?

Lines 83 – 85: Authors should cite a Ref for this phrase.

Line 85: WHO recommendation is this reference? WORLD HEALTH ORGANIZATION - Control of Chagas Disease. Wld Hlth Org. techn. Rep. Ser., (811): 38-47, 1991. Cite this reference.

The citations were worked over as requested. Thank you for the detailed citation information.

Lines 90 – 93: This phrase is not correct because the diagnosis of the acute infection may be performed by serological tests since the first month of infection when IgG is present, associated or not to IgM. Moreover, the acute phase last for approximately four months when IgG is expected to be present in high concentration. PRATA, 2001. RASSI et al., 2010.

According to WHO (WHO technical report Serie 905, Control of CD, Second report of the WHO Expert Committee, 2002, https://apps.who.int/iris/bitstream/handle/10665/42443/?sequence=1) the acute phase lasts 6-8 weeks and then goes over into an indeterminate form, that persists indefinitely. However, it is true, that IgG can be present in the acute phase too, but it is also true that the antibodies need some weeks to be produced and stay positive for years, decades or even lifelong. Therefore it is difficult to assess the state just by looking at IgG.

The definition of acute infections is: that the parasite is found in the blood as a direct proof, as in the acute phase the levels or parasitemia are high. This can be done via microscopy, but this method depends on the experience of the examiner and cannot exclude false positive results by confusion with T. rangeli. To overcome all these problems we added to the serology the NDO-PCR, which is highly sensitive and specific and also a direct proof of T. cruzi in the blood. It is able to detect very small amounts of T. cruzi DNA. So if the PCR does not detect any T. cruzi DNA any more, the chronic phase has started, which is defined according to WHO as a phase where “parasites fall to undetectable levels…”.

However, to address your concerns, we have rephrased the section: “As two different serological techniques are recommended by the WHO and demanded by the Colombian guidelines to entitle the patient for treatment, we applied two different ELISAs. In addition, we performed a Chagas antibody-specific RT. Accordingly to the results, patients being positive in the two ELISAs (and the RT) and negative in all PCR runs were classified as chronically infected and those being positive in the two ELISAs (and the RT) and positive in at least one PCR run as acutely re-infected, re-activated, and/or being in an (early) chronic stage/late acute state of CD infection.”

The negativation of the serological tests in patients infected with T. cruzi and treated occurs when they are cured (WHO, 2002). This process is slow (several years, decades in chronic infections), but occurs. Besides, the progressive drop of the serological tests after treatment is indicative of therapeutic efficacy (Rassi and Luquetti et al.; 2003). Rassi, A., Luquetti AO. 2003. Specific treatment for Trypanosoma cruzi infection (Chagas disease). In: American trypanosomiasis. Tyler, K.M., Miles, M.A., Kluwer Academic Publishers, Boston, USA. pp. 117-125.

See these references and cut or change this phrase.

VIOTTI et al., 2011; Sguassero et al., 2015; 2018; Krettli , 2009;

In your citations and also in many others, a 10% drop after 48 months is seen as a success of treatment, but mostly after long term evaluation a decrease of 20-45% really counts as a therapy success/cure. In our study, we performed ELISAs before and after treatment and found no differences in the titers. As we found no negativation after treatment, we did not include this point, as follow ups were so far not performed. However, we could see with the help of the NDO-PCR a decrease of parasite load of previously positive patients with time and we will publish this data soon.

Materials and Methods

Line 143: Authors should write what want to say what “drug observed treatment” exactly means. This is an error of expression?

As this was also a concern of another reviewer, we rephrased this paragraph: “All volunteers were informed about their results by a physician and registered in the official Colombian data base of the health care provider Dusakawi, guaranteeing all positively tested patients access to treatment. In addition, within the program, there was the offer for all positively tested patients to receive a drug observed treatment, meaning, that the treatment was accompanied for the full term of 2 months by a physician. During the treatment phase all volunteers received physical exams and blood withdrawals to control possible side effects (liver, kidney values, etc.) on a regular base (day 0,7,30,60). Women in childbearing age repetitively received pregnancy tests. In addition, all other occurring complaints, if side effects or not, were treated and documented. A further follow up on the patients is planned.“

Lines 196 – 197: The Disease Classification used by the authors is different than the classic literature that considers an acute case the presence of the parasite detected by the examination of the blood sample in the coverslip observed in 40X microscope objective, associated or not to IgM presence in the serological test plus symptomatology such the signs of parasite T. cruzi entry besides several others not specific of CD (Luquetti, AO and Schmunis, GA Diagnosis of Trypanosoma cruzi infection. In: Telleria Jenny and Michel Tibayrenc. (Org.). American Trypanosomiasis Chagas Disease One Hundred Years of Research. 2ed.Oxford: ELSEVIER, 2017, p. 687-722).

Please see comment above

Again this phrase:  “The last group was interpreted as acutely re-infected, re-activated, and/or as an (early) chronic stage of CD infection.” is different than the classic literature because there is numerous publications that show PCR positive in chronic cases of Chagas disease even after etiological treatment, including in other regions or countries where TcI DTU of T. cruzi is present, as happens in Mexico.  Differently, in regions where TcII, TcV and TcVI the positivity of PCR, is higher as well as the virulence of the parasites and positivity of other parasitological methods of diagnosis.

I understand that this Disease Classification is based on the high credibility of the authors in the RT-PCR performance employed in this study. However, this option needs to be clearly and briefly explained and justified. 

Thank you for this hints. We added in the discussion part the following clarification: ” It needs to be mentioned that cases being positive in PCR and serology could also be classified as acute cases, infected with certain Tc-DTUs that consist even after treatment. Therefore it would have been favorable, if a sequencing could have been done, however, due to limitations of the budget, this was not possible within this study.“

Results

Table 1: % of line 8 of this Table of 8.55 must be changed for 8.56.

Thank you, this was corrected.

Line 254: 8.55% must be changed for 8.56.

Sorry for these typos. As requested, 8.55% has been replaced by 8.56% in the text.

Table 2: Take care with the format, particularly of the columns. Signalize with different markers or symbols the statistically significant differences.

We have to apologize, that we use German nomenclature for tables; we now change these to classical "0" to do not confuse the reader. Second, we perform a Chi2-test in the presented 4x2-table, for which we give the "general p-value" as well as for the test in the 2x2-table. This was mentioned directly in the text beyond the table. To be more precise here we therefore added "general".

Table 4: The alignment of the columns should be ameliorated. I suggest yet the inclusion of a column with the addition of the percentage of reactive cases (serop NDO-neg + serop NDO-pos) for showing easily the situation of all villages.

We respectfully disagree. We have explicitly abstained from providing combined columns, because those would imply overlapping with other columns and thus create more confusion than clearness from our point of view. The two mentioned groups “seropos. NDO-neg.” and “seropos. NDO-pos.” are printed directly next to each other, making it easy for the reader to add them. Due to these reasons, we respectfully ask the editor to accommodate our decision to leave this table as it is.

Discussion

Lines 352 – 355: The cross-reaction of kDNA-PCR between T. cruzi and T. rangeli as well as with Leishmania sp is not a problem today because there are specific primers that avoid this situation. The critical limitation of this procedure is that the kDNA is not quantified and consequently the parasite load is not determined.

We have to respectfully disagree here. Many companies and also institutions use the kDNA-PCR and have cross-reactions with T. rangeli, leading to false positive results. To our knowledge, there is presently no PCR specifically detecting just T. rangeli. Therefore, the NDO-PCR is so valuable from our point of view, as it does not cross-react with T. rangeli as shown in a recent evaluation study. We agree, that Leishmania spp. are not the big trouble in the study area, but as Schijman and Qvarnström have published, the kDNA PCR assessed by these authors also interacts with it.

However we added a sentence in the discussion about the missing quantification of the kDNA, it is now: “Above the kDNA-PCR is limited as quantification is often not possible.“

Conclusions

Lines 366-367:  The phrase “Countermeasures to 366 avoid the further spread of the disease and to improve the situation are urgently needed” is not a conclusion, but an advertisement or recommendation that should be included in the final of the Discussion section.

We transferred the sentence into the discussion section.

Language:

A careful English revision is necessary.

There are some edition errors. Ex. favorale that needs to be replaced by favorable. Some words were written in Spanish.

We worked over the English language and hope, that we have deleted now all typing mistakes.